# Mean Shearing Stroke Frequency of Orthodontic Brackets under Cycling Loading: An In Vitro Study

**DOI:** 10.3390/ma13194280

**Published:** 2020-09-25

**Authors:** Orhan Cicek, Nurhat Ozkalayci, Mehmet Yetmez

**Affiliations:** 1Department of Orthodontics, Faculty of Dentistry, Zonguldak Bulent Ecevit University, 67600 Zonguldak, Turkey; nurhat.ozkalayci@beun.edu.tr; 2Department of Mechanical Engineering, Faculty of Engineering, Zonguldak Bulent Ecevit University, 67100 Zonguldak, Turkey; yetmez@beun.edu.tr

**Keywords:** orthodontics, orthodontic bracket, acid-etching, self-etching, bonding, cycling loading, stroke

## Abstract

Based on the development of many adhesive systems and bonding techniques, bonding strength of orthodontic brackets has become even more important in modern clinical orthodontics. The aim of this study was to determine mean shearing stroke frequency of different orthodontic bracket types and bonding agents under cycling loading. Therefore, 10 different types of orthodontic bracket from 4 different brands were divided into 2 groups. Two different adhesives, namely Transbond™ XT etch-and-rinse for *Group 1* and Transbond™ Plus self-etching-primer adhesive for *Group 2* were considered. The brackets were tested under cycling loading force of 10-N and a crosshead speed of 300 mm/min and 40 cycle/min. The frequency of strokes that the brackets failed were determined and these data were analyzed by statistical analysis using an independent sample *t*-test and one-way analysis of variance (ANOVA). The level of significance was set at *p* < 0.05. Generally, differences between the frequency of shearing strokes of the bracket failures were found to be statistically significant depending on the type of adhesives and brackets (*p* < 0.05). The bonding technique for Group 1 was found to have a significantly higher shear bonding strength than Group 2. It is also seen that different types of bracket belonging to the same or different brands had different shear bonding strength. It may be concluded that: (i) all bracket types used in this study can be applied with both bonding techniques, (ii) in order to minimize the risk of hard tissue damage, ceramic brackets should be carefully bonded using the self-etching primary adhesive technique.

## 1. Introduction

Modern orthodontic developments have focused on lessening orthodontic treatment time and achieving the perfect and stable treatment outcomes [1,2]. Bracket failures that delay the achievement of these outcomes may occur due to various factors, from occlusal trauma to inappropriate bonding techniques [3,4]. However, patient compliance is a very important factor in terms of treatment time and outcomes. Since incompatible patients do not follow the instructions exactly, they risk the outcome of the treatment by increasing the probability of failure of orthodontic attachments [3,5,6].

The bonding of orthodontic attachments is one of the most important phases of the entire treatment process. The bonding technique can be applied incorrectly not only by an inexperienced clinician, but also by an experienced orthodontist who does not follow the procedures carefully [7]. Success in bonding strength is achieved by understanding and sticking to accepted orthodontic and protective dentistry principles such as cleaning the adhesive surfaces, ensuring appropriate wetting and firm adaptation, and achieving sufficient polymerization [8].

Advances in orthodontic adhesive and bracket base manufacturing have created many options for clinicians, and some combinations have been reported to perform best [9,10]. It has been reported that the bracket base design differences are one of the main factors that may affect the adhesive penetration [9,11,12]. The bracket bonding strength is directly related to the polymerization of the adhesive under the bracket. This polymerization is affected by the amount of light that penetrates the adhesive rather than the amount of light scattered. The high bonding strength reduces bending moments in the bracket/adhesive system that is exposed to heavy chewing forces throughout the treatment process [13].

Bonding failure of orthodontic brackets is an undesired situation for orthodontists and patients, causing increased appointments, prolonged chair time and relapse in tooth movements [14]. During orthodontic treatment, the bonding strength of the brackets should be at a level that will provide adequate adhesive strength against the transfer of orthodontic force to the tooth, biting forces, and also allow it to remove without damaging the tooth during the finishing phase. Although vertical occlusal bite forces in humans vary between 7–16 kg on average, this can go up to 15–30 kg at maximum bite [15]. The clinically admissible bracket bonding strength level is in the range of 5.9–7.8 MPa [16]. Undesired bracket bonding failure rates are from 0.6% to 6.6% [17,18]. Undesired bracket failures during fixed orthodontic treatment depend on enamel surface properties, bracket type, physician’s technique and patient compliance [17,18,19].

Shear bonding strength tests were first applied in the late 1970s and are widely used in measuring the bonding strength of orthodontic materials [20]. Shear forces include shear and peel forces applied at a distance from the adhesive interface [21,22]. Shear bonding strength tests are the most widely used laboratory method applied using various techniques to measure the bonding strength of orthodontic attachments [23].

The aim of this study was to determine mean shearing stroke frequency of different orthodontic bracket types and bonding agents under cycling loading.

## 2. Material and Methods

### 2.1. Ethical Approval and Preparation of Samples

Ethics committee approval for the study was given by the Clinical Research Ethics Committee at Zonguldak Bulent Ecevit University (2016/05: 2016-21-27/01). Three hundred upper premolar human teeth were divided into two subgroups with 15 samples. The sample size was determined using the data from Dall’Igna et al. [24]. A sample size of 15 teeth per subgroup would give more than 95% power to detect significance differences at the 0.05 level. Teeth such as no-tooth with filling or restoration/coloration, no-tooth with fracture or crack on the enamel surface, no-tooth with a history of tooth bleaching or/and malformation on the buccal surface were included [25]. Teeth possessing at least one of these in the anamnesis were excluded from the study.

### 2.2. Storage Conditions of Teeth and Preparation of Acrylic Blocks

The teeth were kept in distilled water with 0.1% thymol crystals to prevent dehydration and deterioration of the enamel structure after removing blood, organic residue, etc. [26]. Teeth were embedded in acrylic blocks from the cement–enamel junction. Then, the samples were kept in distilled water for 24 h to prevent drying prior to testing.

### 2.3. Brackets Used

The following 10 different types of upper premolar brackets were considered in this study (see Figure 1 and Figure 2):

(*a*) APC (Adhesive Pre-coated) Victory metal (3M Unitek; Monrovia, CA, USA),

(*b*) Victory metal (3M Unitek; Monrovia, CA, USA),

(*c*) Gemini metal (3M Unitek; Monrovia, CA, USA),

(*d*) Gemini Clear ceramic (3M Unitek; Monrovia, CA, USA),

(*e*) Gemini SL self-ligating metal (3M Unitek; Monrovia, CA, USA),

(*f*) Clarity Advanced ceramic (3M Unitek; Monrovia, CA, USA),

*(g*) Clarity SL self-ligating ceramic (3M Unitek; Monrovia, CA, USA), 

(*h*) Equilibrium^®^ metal (Dentaurum, Inspringen, Germany), 

(*j*) Mini Sprint^®^ metal (Forestadent Company, Pforzheim, Germany), 

(*k*) Mini Master metal (American Orthodontics, Sheboygan, NY, USA).

### 2.4. Bonding of Brackets

Vestibular surfaces of the teeth were cleaned by fluorine-free pumice-water mixture using polishing rubber, rinsed and dried. According to the information provided by the manufacturer (3M Unitek, Safety Data Sheet), the compositions of the adhesive agents used in the study are summarized in Table 1. In *Group 1 (etch-and-rinse)*, Transbond™ XT Light Cure Adhesive Primer (3M Unitek, Monrovia, CA, USA) and Transbond™ XT Light Cure Adhesive Paste (3M Unitek, Monrovia, CA, USA) were used to bond the brackets. In *Group 2 (self-etching-primer)*, Transbond™ Plus Self Etching Primer (3M Unitek, Monrovia, CA, USA) and Transbond™ XT Light Cure Adhesive Paste (3M Unitek, Monrovia, CA, USA) were used to bond the brackets.

### 2.5. Preparation of Groups

For *Group 1 (etch-and-rinse)*, acid-etching was applied with 37% phosphoric acid for 30 s and then rinsed with water for 30 s and air dried for 10 s. After seeing the chalky white surface, Transbond™ XT Light Cure Adhesive Primer (3M/Unitek, Monrovia, CA, USA) was applied as a thin layer on the tooth surface.

For *Group 2 (self-etching-primer)*, routine care control was completed so that the buccal surfaces of the teeth were clean. Transbond™ Plus Self Etching Primer (3M/Unitek, Monrovia, CA, USA) was applied to the tooth surface for five seconds. Consequently, air was feebly applied to the surface with an air-water syringe. Then, APC brackets were removed from their protective boxes and placed in their correct position on the tooth surface. Transbond™ XT Light Cure Adhesive Paste (3M/Unitek, Monrovia, CA, USA) was placed on the base of the remaining brackets. All brackets were placed in the precise position on the tooth surface and the flashes were removed. All brackets were cured from mesial and distal for 20 s using 3M Espe Elipar S10 (3M ESPE Dental Products) which has the intensity of 430–480 nm at about 1200 mW/cm^2^, respectively.

### 2.6. Fatigue Testing

Fatigue tests were carried out with a closed-loop controlled, low-cycle fatigue machine with 10 N capacity and a crosshead speed of 300 mm/min and 40 cycle/min. During the test, attention was paid to the force coming perpendicular to the bracket groove. After fixing the acrylic blocks, the test was conducted to obtain frequency of strokes according to the bracket failure (see Figure 3).

### 2.7. Statistical Analysis

Statistical analysis was performed using SPSS for Windows (version 23.0; SPSS, Chicago, IL, USA). For comparisons between groups, an independent sample *t*-test and one-way analysis of variance, post-hoc Tukey test and Tamhane’s T2 were chosen. *p* value less than 0.05 was considered statistically significant.

## 3. Results

According to the descriptive statistical analysis of the groups, the homogeneity test results of the groups are shown in Table 2; the statistically significantly difference between groups and subgroups according to the ANOVA test are given in Table 3 and Table 4.

The number of brackets used in the study, the mean frequency of strokes, the standard deviation, the minimum and maximum frequency of strokes are given in Table 5.

### 3.1. Results of APC Brackets

For both *Group 1* and *Group 2*, no statistically significantly difference was found between the mean shearing stroke frequency of APC metal brackets and Victory metal brackets (*p* > 0.05). The shearing stroke frequency was found to be statistically significantly higher in APC metal brackets compared to Mini Sprint^®^ and Mini Master metal brackets (*p* < 0.05).

For *Group 2*, a statistically significantly lower shearing stroke frequency was found in APC metal brackets compared to Equilibrium^®^ 2 metal brackets (*p* < 0.05).

### 3.2. Results According to Bracket Raw Materials

For both *Group 1* and *Group 2*, even though the mean shearing strokes frequency of Clarity Advanced ceramic brackets were higher than APC metal, Victory metal, Equilibrium^®^ 2 metal brackets, no statistically significant differences was found (*p* > 0.05). In spite of the mean frequency of shearing strokes of Clarity Advanced ceramic brackets were higher compared to Clarity SL self-ligating ceramic brackets, no statistically significant difference was found. The mean shearing stroke frequency of Victory series metal brackets was found to be statistically significantly higher than that of Mini Sprint^®^ metal and Mini Master metal brackets (*p* < 0.05).

For *Group 1*, there was no statistically significant difference between Equilibrium 2 metal brackets and Victory series metal brackets in mean shearing stroke frequency (*p* > 0.05). The mean frequency of shearing strokes of Clarity Advanced ceramic brackets was found to be statistically significantly higher except for APC Victory, Victory and Equilibrium^®^ 2 metal braces (*p* < 0.05).

For *Group 2*, a statistically significantly higher mean frequency of shearing strokes was found in Equilibrium 2 metal brackets compared to Victory series metal brackets. Mean shearing strokes frequency was found to be significantly higher than that of Mini Master and Mini Sprint^®^ metal brackets (*p* < 0.05).

### 3.3. Results According to Ligation Types

For both *Group 1* and *Group 2*, mean shearing stroke frequency of Gemini SL self-ligating metal brackets were found to be statistically significantly higher than Mini Master and Mini Sprint^®^ metal brackets. Mean shearing stroke frequency of Equilibrium^®^ 2 metal brackets was found to be statistically significantly higher than Gemini SL self-ligating metal brackets (*p* < 0.05). Additionally, there was no statistically significant difference between the mean frequency of shearing strokes of Clarity SL self-ligating ceramic brackets and Clarity Advanced ceramic brackets (*p* > 0.05).

For *Group 1*, mean shearing stroke frequency of Clarity SL self-ligating ceramic brackets was found to be statistically higher than Gemini SL self-ligating metal brackets (*p* < 0.05). However, for *Group 2*, there was no statistically significant difference between each other (*p* > 0.05).

### 3.4. Results According to Brackets Brand

For both *Group 1* and *Group 2*, mean shearing stroke frequency of 3M Unitek and Dentaurum brand metal brackets were found to be statistically significantly higher than Forestadent and American Orthodontics brand metal brackets (*p* < 0.05). Moreover, there was no statistically significant difference between the shearing strokes frequency of 3M ceramic brackets (*p* > 0.05).

On the one hand, for *Group 1*, there were mostly no statistically significant differences between the mean shearing strokes frequency of 3M brand and Dentaurum brand metal brackets. On the other hand, for *Group 2*, mean shearing strokes frequency of Dentaurum brand brackets were found to be significantly higher than 3M brand metal brackets (*p* < 0.05).

In a similar way, on one hand, for *Group 1*, there were no statistically significant differences between mean shearing strokes frequency of Forestadent brand and American Orthodontics brand metal brackets (*p* > 0.05). On the other hand, for *Group 2*, the mean shearing strokes frequency of American Orthodontics brand metal brackets were found to be statistically higher than Forestadent brand metal brackets (*p* < 0.05). In both groups, Mini Sprint^®^ metal and Mini Master metal brackets showed statistically significantly lower mean shearing stroke frequency, the standard deviation, and the lower minimum and maximum number of strokes (*p* < 0.05).

### 3.5. Results According to Type of Adhesive Agents

Table 6 shows that the mean frequency of strokes of *Group 1* brackets were statistically significantly higher than that of *Group 2* brackets.

### 3.6. Bracket Wing Breakings and Dental Hard Tissue Damages

For Clarity Advanced ceramic brackets, samples were renewed due to breakage in the bracket wings in four samples and severe hard tissue damage (enamel + dentin) in two samples (see Figure 4). Most of the Gemini Clear ceramic brackets were excluded from the statistical evaluation as the bracket wing broke during the cycling testing (see Figure 5).

## 4. Discussion

Shearing forces are applied to orthodontic brackets mostly in shear bonding strength tests [27]. Although various types of blade are used in the shear bonding strength tests, it is seen that most of the studies used scissors blades and in some studies a wire loop [28,29]. In this study, since the use of wire loops creates tensile stresses, a nozzle head specially made of stainless steel was used to load shear forces and minimize tensile stresses [30].

The factors affecting in vitro orthodontic bonding strength testing were reported as: test conditions, storage conditions of the samples which were embedded to blocks, polymerization time, and crosshead speed [31]. However, many other variables can alter the bonding strength of orthodontic brackets, such as temperature, fluoride content or filler size [32].

In orthodontic bracket bonding strength tests, shearing forces are applied on the bracket base, ligature groove or bracket wings. These three different locations where shearing forces are applied display significant differences between studies. Klocke at al. reported a 49.3% decrease in SBS values and 25% increase in bonding failure when the shearing force was transferred from the bracket–resin interface to the ligature groove [33]. Thomas et al. reported a 48% difference between the shear bonding strength occurring at a distance of 300 µm from the enamel surface and the enamel surface [34]. In this study, shearing forces were applied vertically on the bracket groove in the inciso-gingival direction.

Bonetti et al. reported that there was no statistically significant difference in their study comparing the bonding failure mode of APC II Victory (3M Unitek) series metal brackets and Victory metal (3M Unitek) brackets [35]. Abd and Al-Khatieeb reported that the shear bonding strength of APC™ Flash-Free brackets and Victory series low-profile metal brackets were similar and no statistically significant difference was found [36]. The results of present study were consistent with the studies in the literature; for both *Group 1* and *Group 2*, mean shearing strokes frequency of APC series metal brackets and Victory series metal brackets were similar and no statistically significantly difference was found. On one hand, APC Victory metal brackets in *Group 1* and *Group 2* had a statistically significantly higher shearing stroke frequency when compared to Mini Sprint^®^ metal and Mini Master metal brackets. On the other hand, APC metal brackets in *Group 2* possessed a statistically significantly lower shearing stroke frequency than that of Equilibrium^®^ 2 metal brackets.

Mirzakouchaki et al. reported in their study that the bonding strength of metal brackets (Roth 022, Ovation, GAC International, Inc.) was statistically significantly higher than ceramic brackets (Roth 022, Allure III, GAC International, Inc.) [37]. Liu et al. reported that there was no statistically significant difference between the shear bonding strength of ceramic brackets (Clarity, 3M Unitek) and conventional metal brackets (Tomy, Tokyo, Japan) [38]. However, Bishara et al. reported that metal brackets (Victory, 3M Unitek) had a statistically significantly lower shear bonding strength than ceramic brackets (Clarity, 3M Unitek) [39]. These differences between studies may be attributed to the different adhesive agents used, the method and time of force application, and the types of bracket. In this study, in accordance with the different findings in the literature, the frequency of shearing strokes of ceramic brackets in both *Group 1* and *Group 2* were evaluated as higher than metal brackets. In *Group 1*, Clarity Advanced ceramic brackets had statistically significantly higher shearing stroke frequency compared to Mini Master metal, Mini Sprint^®^ metal, Gemini metal and Gemini SL self-ligating metal brackets. In *Group 2*, a statistically significantly higher shearing stroke frequency was found in Clarity Advanced ceramic brackets compared to Mini Master metal and Mini Sprint^®^ metal brackets. Although the frequency of shearing strokes was noted to be higher than the remaining metal brackets in both *Group 1* and *Group 2*, no statistically significantly difference was found.

Sfondrini et al. reported in their study that the shear bonding strength of self-ligating metal brackets (Smart Clip^®^, 3M Unitek) was statistically significantly higher than conventional metal brackets (Step^®^; Leone) [40]. Northrup et al. reported that although both brackets showed adequate clinically acceptable bond strength, the shear bonding strength of self-ligating metal brackets (Damon 2, Ormco Corporation) was statistically significantly higher than conventional metal brackets (Orthos, Ormco Corporation, Glendora, Calif) [41]. For both *Group 1* and *Group 2*, a statistically significantly higher shearing stroke frequency was observed in Gemini SL self-ligating metal brackets compared to Mini Master metal and Mini Sprint^®^ metal brackets. Additionally, a statistically significantly higher shearing stroke frequency was calculated in Equilibrium^®^ 2 metal brackets compared to Gemini SL self-ligating metal brackets.

Theodorakopoulou et al., in their in vitro study comparing the shear bonding strength of polycrystalline (Clarity, 3M Unitek) and monocrystalline (Inspire, Ormco) ceramic brackets, reported that the bonding strength was similar in both ceramic bracket types and no statistically significant differences were found [42]. In this study, the shearing stroke frequency of the Clarity SL self-ligating ceramic brackets in both *Group 1* and *Group 2* was similar to that of the Clarity Advanced ceramic brackets, and no statistically significant difference was found.

Zielinski et al. reported that there were 8 bracket breakages in different types of ceramic and plastic bracket groups investigating the shear bonding strength [43]. Stein et al. worked on the effect of 445 nm diode laser application on enamel damage and bonding failure and reported that there were breakages in two samples in the group using conventional bracket removal pliers [44]. Similar to previous studies, samples of our study were renewed in Clarity Advanced ceramic brackets due to breakage in the bracket wings in four samples during the test. However, in the most of Gemini Clear ceramic brackets, which were thought to be incapable of being reliably statistically analyzed, breakage was encountered in the wings during the test.

Jeiroudi reported enamel damage in the upper right central and lateral incisors which was broken due to an accident in the patient and treated using a polycrystalline sapphire ceramic bracket [45]. Viazis et al. reported that in their in vitro study in which they investigated the shear bonding strength and potential enamel damage, they encountered fracture involving enamel and dentine up to the pulp horn in only one sample in the group using polycrystalline ceramic brackets [46]. In this study, severe fracture was encountered in two samples of Clarity Advanced ceramic brackets involving enamel and dentine.

Sheibaninia et al. reported that Equilibrium 2 metal brackets showed the statistically significantly highest bonding strength in distilled water compared to self-ligating metal brackets, while a statistically significantly decrease in bonding strength in acetic acid medium [47]. Similarly, Cozza et al. reported that if the values were specified in newton, Victory series and Equilibrium 2 metal brackets showed the highest bonding strength. In addition, they concluded that the bonding strength of Victory series metal brackets with both Mini Sprint and Equilibrium 2 metal brackets were similar and there was no statistically significant difference [48].

In this study, it was clearly seen for both *Group 1* and *Group 2* that:(i)statistically significantly higher shearing strokes frequency was found in Victory series metal brackets when compared with Mini Sprint^®^ metal and Mini Master metal brackets,(ii)equilibrium 2 metal brackets showed statistically significantly higher shearing stroke frequency than that of Gemini SL self-ligating metal brackets.

Bishara et al., in their in vitro study investigating the bonding strength using metal brackets (Victory series, 3M Unitek), reported statistically significantly higher bonding strength in the acid-etching group compared to the self-etching group [49]. Türk et al., concluded that there was no statistically significant difference between conventional acid etching or self-etching in bonding strength after 24 h in their study [50]. Yamada et al. reported that the phosphoric acid etching group possessed statistically significantly higher bonding strength compared to the other groups [51]. In this study, the shearing stroke frequency of the brackets for *Group 1* were computed to be statistically significantly higher than that of self-etching brackets for *Group 2*.

Farahani et al., reported that bonding strength was statistically significantly higher in 3M brand brackets (Gemini metal brackets) and significantly lower in American Orthodontics brand brackets (i.e., Master series metal brackets) in the control group without aluminum oxide sandblasting. There was no statistically significant difference between American Orthodontics and Dentaurum brand brackets (Discovery metal bracket) [52]. Dalaie et al. reported that the highest statistically significant bonding strength was found in Dentaurum and American Orthodontics brand metal brackets and there was no statistically significant difference between Dentaurum and American Orthodontics brand metal brackets [53].

In this study, the shearing strokes frequency of 3M Unitek and Dentaurum brand metal brackets were found to be statistically significantly higher than those of Forestadent and American Orthodontics brand metal brackets. Dentaurum brand brackets were found to be statistically significantly higher shearing strokes frequency than that of 3M brand metal brackets in *Group 2*. In both *Group 1* and *Group 2*, breakage was observed at statistically significantly lower frequency of strokes in Forestadent and American Orthodontics brand metal brackets among all brands. In addition, difference between the shearing stroke frequency of Forestadent and American Orthodontics brand metal brackets were not found to be statistically significant in *Group 1*, while in *Group 2*, frequency of shearing strokes of American Orthodontics brand metal brackets were statistically significantly higher than Forestadent brand metal brackets. There was no statistically significant difference between 3M branded ceramic brackets. In both *Group 1* and *Group 2*, there was no statistically significant difference found between Dentaurum branded metal brackets and 3M branded ceramic brackets.

Although it is actually obvious that this study does not totally reflect the complexity of the in vivo conditions, it presents a novel approach on evaluation of shear bonding strength of brackets with a different method in which this is tested under cyclic loading. However, another important limitation of the study is only one adhesive agent brand (3M Unitek) is used [54]. The reason for preferring this novel method is that cyclic loading mechanism of the fatigue machine used in the study is similar to vertical cyclic forces exerted on teeth by physiological biting movements.

## 5. Conclusions

For the sake of planning according to time span, frequency of shearing strokes plays an important role for understanding orthodontic brackets performance.

As per the primary results of the study, it is suggested that all types of brackets can be bonded with both adhesive agents. However, it was underlined that in order to reduce the risk of hard tissue damage, ceramic brackets could be bonded with the self-etching primary technique carefully. In spite of same type of bonding agent being used between different brands of metal bracket, there were differences in mean shearing stroke frequency. The reason for this difference was thought to be due to different techniques and designs used in the production of brackets.

As per the secondary results of the study, independently of the adhesive agent (etch-and-rinse or self-etching primer), Mini Sprint^®^ metal and Mini Master metal brackets showed lower mean shearing stroke frequency, standard deviation, and lower minimum and maximum frequency of strokes. In addition, at least one of the bracket wings of all Gemini Clear ceramic brackets was broken during the fatigue tests. However, Equilibrium^®^ 2 metal and Victory metal brackets showed higher mean shearing stroke frequency with both bonding agents.

Considering the limitations arising from this in vitro study, the effectiveness of fatigue tests should be supported by advanced in vitro studies.

## Figures and Tables

**Figure 1 materials-13-04280-f001:**
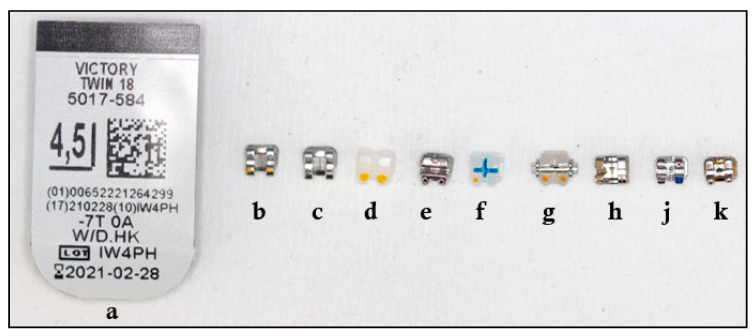
Brackets used in the study.

**Figure 2 materials-13-04280-f002:**
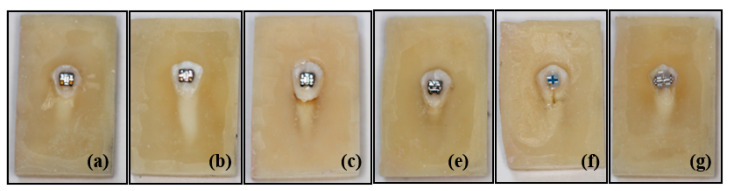
Photos of some samples of APC, metal, ceramic and self-ligating brackets used in the study. (**a**) APC Victory metal, (**b**) Victory metal, (**c**) Gemini metal, (**e**) Gemini SL self-ligating metal, (**f**) Clarity Advanced ceramic, (**g**) Clarity SL self-ligating ceramic bracket.

**Figure 3 materials-13-04280-f003:**
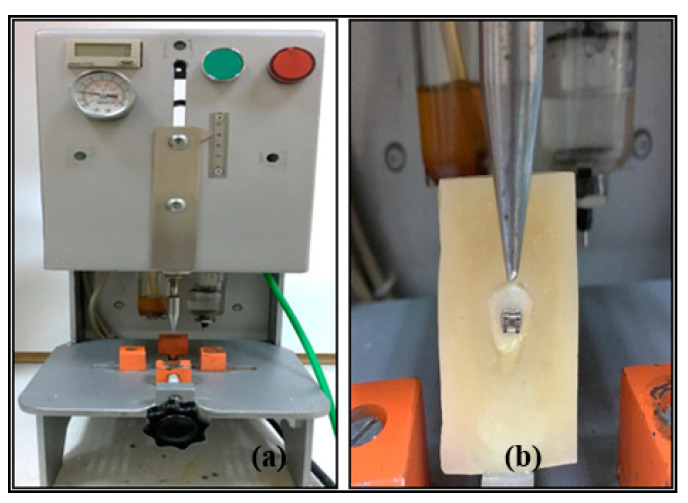
(**a**) A closed-loop controlled low-cycle fatigue machine, (**b**) direction of shearing force.

**Figure 4 materials-13-04280-f004:**
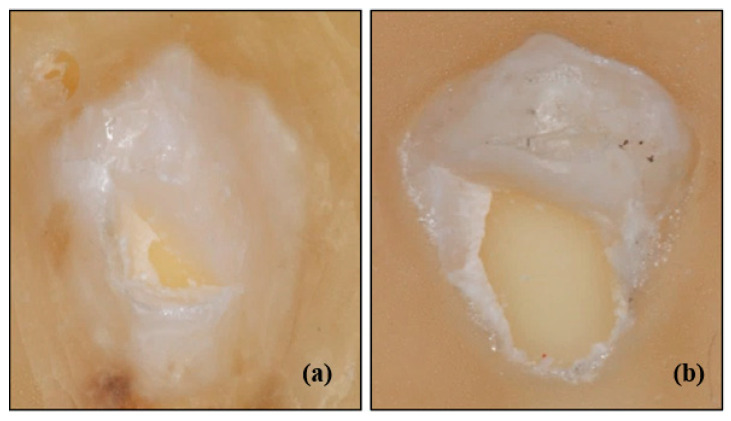
(**a**) Clarity Advanced ceramic brackets with hard tissue (enamel + dentin) damage, (**b**) Clarity Advanced ceramic brackets with severe hard tissue (enamel + dentin) damage.

**Figure 5 materials-13-04280-f005:**
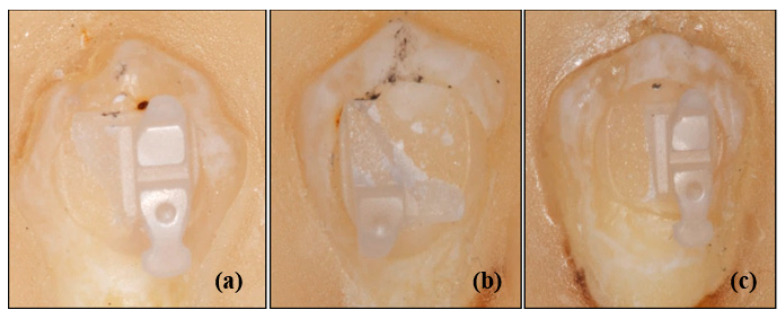
Gemini Clear ceramic bracket samples with wing break. (**a**,**c**) Two of the four bracket wings were broken, and two intact bracket wings remained, (**b**) Three of the four bracket wings were broken, and one intact bracket wing remained.

**Table 1 materials-13-04280-t001:** Description and composition of adhesive agents used in the study.

Manufacturer	Adhesive Materials	Group/Groups Used	Composition of Adhesive Agent	% by Wt
			Bisphenol A diglycidyl ether dimethacrylate (BISGMA)	45–55
3M Unitek, Monrovia, CA, USA	Transbond™ XT Light Cure Adhesive Primer	Group 1 (etch-and-rinse)	Triethylene glycol dimethacrylate (TEGDMA)	45–55
			4-(Dimethylamino)-benzeneethanol	<0.5
			Methacrylated pyrophosphates	10–25
			Ethylene dimethacrylate	0–2
			Phosphoric acid	0–2
3M Unitek, Monrovia, CA, USA	Transbond™ Plus Self Etching Primer	Group 2 (self-etching-primer)	2-Hydroxyethyl methacrylate (HEMA)	<1
			2-Propenoic acid, 2-methyl-, phosphinicobis (oxy-2,1-ethandiyl) ester	25–40
			Water	15–25
			DL-Camphorquinone	<3
			Silane-treated quartz	70–80
			Bisphenol A diglycidyl ether dimethacrylate (BISGMA)	10–20
			Bisphenol A dimethacrylate	5–10
3M Unitek, Monrovia, CA, USA	Transbond™ XT Light Cure Adhesive Paste	Group 1 and Group 2	Silane-treated silica	<2
			Diphenyliodonium hexafluorophosphate	<1
			Triphenylantimony	<1

**Table 2 materials-13-04280-t002:** Homogeneity test results of the groups.

Groups	Levene Statistics	df1	df2	Significance Value (*p*)
*Group 1*	1.56	9	140	0.13
*Group 2*	3.32	9	140	0.001 *

*: Means equal variances not assumed at *p* < 0.05.

**Table 3 materials-13-04280-t003:** Statistical comparison of Group 1 means.

**Group**		Sum of Squares	df	Mean of Squares	F	Significance Value (*p*)
*Group 1*	Between groups	4651.57	9	516.84	28.62	<0.001
	In-group	2528.26	140	18.05		
	Total	7179.84	149			

**Table 4 materials-13-04280-t004:** Statistical comparison of Group 2 means.

**Group**		Sum of Squares	df	Mean of Squares	F	Significance Value (*p*)
*Group 2*	Between groups	2853.62	9	317.07	35.54	<0.001
	In-group	1249.06	140	8.92		
	Total	4102.69	149			

**Table 5 materials-13-04280-t005:** Values of shearing strokes (S) of brackets.

Group	Bracket Type	Sample	Strokes	S_min_	S_max_
*Group 1*	APC Victory series metal bracket	15	18.00 ± 4.79	9	28
	Victory series metal bracket	15	19.06 ± 3.89	12	25
	Gemini metal bracket	15	15.26 ± 4.49	5	22
	Gemini Clear ceramic bracket	15	24.46 ± 4.24	19	32
	Gemini SL self-ligating metal bracket	15	14.66 ± 2.76	10	19
	Clarity Advanced ceramic bracket	15	20.86 ± 3.48	12	26
	Clarity SL self-ligating ceramic bracket	15	22.26 ± 7.28	14	45
	Equilibrium^®^ 2 metal bracket	15	19.66 ± 4.65	13	33
	Mini Sprint^®^ metal bracket	15	6.40 ± 2.22	4	12
	Mini Master series metal bracket	15	8.13 ± 2.16	4	11
	**Mean**	**150**	**16.88 ± 6.94**	**4**	**45**
*Group 2*	APC Victory series metal bracket	15	12.40 ± 2.64	10	18
	Victory series metal bracket	15	13.73 ± 3.97	8	22
	Gemini metal bracket	15	12.33 ± 2.19	8	16
	Gemini Clear ceramic bracket	15	20.86 ± 4.18	15	29
	Gemini SL self-ligating metal bracket	15	12.26 ± 2.54	9	18
	Clarity Advanced ceramic bracket	15	16.13 ± 3.48	10	23
	Clarity SL self-ligating ceramic bracket	15	15.06 ± 3.69	10	22
	Equilibrium^®^ 2 metal bracket	15	19.20 ± 2.78	14	25
	Mini Sprint^®^ metal bracket	15	5.86 ± 1.68	4	9
	Mini Master series metal bracket	15	7.86 ± 1.12	6	9
	**Mean**	**150**	**13.57 ± 5.24**	**4**	**29**

**Table 6 materials-13-04280-t006:** Comparison of the mean frequency of shearing strokes of the groups with adhesive agent.

Adhesive Agent	Sample Size	Strokes
*Group 1* (Transbond XT)	150	16.88 ± 6.94
*Group 2* (Transbond Plus)	150	13.57 ± 5.24

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
