# Peer review of "Mean Shearing Stroke Frequency of Orthodontic Brackets under Cycling Loading: An In Vitro Study"

_materials, 2020, doi:10.3390/ma13194280_

Round 1
Reviewer 1 Report
Different types of brackets belonging to the same or different brands had different shear bonding behaviour. All bracket types used in this study could be applied with both bonding techniques. To minimize the risk of hard tissue damages, ceramic brackets should be carefully bonded using the self-etching primary adhesive technique.
Strengths: Design is adequate.
Weaknesses: Additional English editing should be done to make it more readible.
Abstract: "Bonding technique for Group 1 says that..." Please rephrase as "Bonding technique for Group 1 was found to have a significantly higher shear bonding than Group 2."
Tables 1-3: heading of first column should be replaced. For Tables 2-3, please show exact p values. If they are very small, then write p < 0.001. For Tables 2-3, the legend should be statistical comparison, not statistically.
Table 5: is the MSSN the same meaning as # of strokes in Table 4? If yes, please unify them.
Please send for English editing.
Reviewer 2 Report
TITLE: please indicate the type of study
ABSTRACT:
You should specify that different types of brackets are tested.
page 1, line 11: this sentence should be removed.
page 1, lines 10-14: the verb tense is not consistent. Present simple in lines 10-14, past simple in lines 15-16.
page 1, lines 28-37: this sentence should be removed
page 1, lines 39-42: the meaning of this sentence is not clearly expressed; please try to rephrase this sentence
page 1, line 43: " on average" instead of "of mean"
page 2, line 45: a reference is missing there "rates are from 0.6% to 6.6%."
MATERIALS AND METHODS:
page 2, lines 64-69: the companies are not specified
page 2, line 77: "vestibular" instead of "vestibule"
page 3, line 84: "rinsed" instead of "washed"
RESULTS:
page 5: lines 119-120: please try to rephrase this sentence, as you repeat twice "Victory series metal brackets."
DISCUSSION:
please add a paragraph with "limitations and strenghts of the study"
the sample size calculation has been performed?
Dear authors, the article need a professional language editing service before acceptance.
Reviewer 3 Report
The paper entitled "Bonding Behaviour of Orthodontic Brackets Under3 Cycling Loading" is an interesting contribute that evaluate
mechanical bonding behaviour of different orthodontic bracket
types and bonding agents under cycling loading.
However, some corrections are needed before the paper
can be considered suitable for publication
Introduction
Introduction section is too brief, it highlights only the more general
characteristics of the adhesion of the orotodontic bracket.
The various factors that may affect this adhesion are not mentioned,
nor those related to the characteristics of the bracket,
nor those that depend on the adhesive technique.
The bibliography is relevant, however this section must be
largely revised.
Material and Methods
Material and methods section is reported in a clear and
detailed manner. All the different steps of the study are described
analytically and are easily understood by the reader.
In the present study only metallic and ceramic brackets were tested,
for which reason sapphire brackets have not been evaluated,
considering their extensive use in the clinical practice.
Authors must also clarify which type of brackets were tested,
whether incisors, canines or premolars
Results The results of the study are overall well represented also from an
iconographic point of view. The statistical analysis was carried out
correctly and allows a reliable evaluation of the results obtained;
the representation in the table of the statistical data is more
clear and analytical
Discussion The literature analysis reported in the discussion section is too
extensive, it is necessary to review this part of the work and
better analyze the data from the work
Reviewer 4 Report
- It is necessary to add the number of cycles. You describe the load and crosspeed, but not how many times did. - In table 1, what does the asterisk next to p value in group 2 mean? It should be explained in a caption under the table. - It would also be convenient to put photos of how the metal brackets were, even if it is only a group, to be able to see the difference with the ceramic brackets. - In line 87 and 88, it refers to the seconds of use of the primer, the manufacturer's indications, are at least 3 to 5 seconds, and here it puts a vague description, which could be even less than recommended, "approximately 3 seconds", should be more concrete. - Of the 34 bibliographic references, only 9 are from 2010 or higher, the subject of bonding is changing a lot, it would be convenient to update it
Specifically, line 44 refers to the strength of adhesion, by means of a bibliographic citation from 1975, it should be updated.
Round 2
Reviewer 1 Report
The authors have satisfactorily addressed my concerns.
Reviewer 2 Report
Dear authors,
I think that the article has been significantly improved.
Please find some final suggestions:
TITLE: I would suggest to change "Bonding behaviour" with "mean shearing stroke frequency"
INTRODUCTION:
- in the introduction, I would suggest to add a definition of "Shearing forces" and "shear bonding strength tests"
- page 1, line 50: please add an updated reference to this sentence; as a suggestion, please find a reference below:
Staderini E, Guglielmi F, Cornelis MA, Cattaneo PM. Three-dimensional prediction of roots position through cone-beam computed tomography scans-digital model superimposition: A novel method. Orthod Craniofac Res. 2019;22(1):16-23. - page 2, line 67: please add some updated references to this sentence, as cleft lip patients are typical patients with tooth malformation; as a suggestion, please find the references below:
- Staderini E, De Luca M, Candida E, et al. Lay People Esthetic Evaluation of Primary Surgical Repair on Three-Dimensional Images of Cleft Lip and Palate Patients. Medicina (Kaunas). 2019;55(9):576. Published 2019 Sep 8. doi:10.3390/medicina55090576
- My suggestion is to xlearly specify primary and secondary outcomes of this study: "Which is the mean shearing stroke frequency of commercially available brackets?"
- As a suggestion, I would use "mean shearing stroke frequecy" instead of "mean shearing stroke number"
MATERIALS AND METHODS:
can you specify which is the brand of the curing light?
You can change "Group 1" and "Group 2" with "etch-and-rinse" or "self-etching primer"
RESULTS:
you can say that, in both groups, Mini Sprint® metal bracket and Mini Master series metal bracket showed lower mean shearing stroke frequecy, the standard deviation, and the lower minimum and maximum number of strokes.
DISCUSSION:
In the limitation section, can you specify how your method can be defined "different? Could you add the reason of this choice?
Moreover, as a limitation, you can add that only one bonding brand (3M) was used. As a suggestion, please add the reference below: Patini R, Staderini E, Camodeca A, Guglielmi F, Gallenzi P. Case Reports in Pediatric Dentistry Journals: A Systematic Review about Their Effect on Impact Factor and Future Investigations. Dent J (Basel). 2019;7(4):103. Published 2019 Oct 24. doi:10.3390/dj7040103
CONCLUSION:
The conclusion should be consisten with the topic. The question is reported in the comments above.
If I correctly understoood, you could say that, indipendently of the bonding type (etch-and-rinse or self-etching primer) Mini Sprint® metal bracket and Mini Master series metal bracket showed lower mean shearing stroke frequecy, the standard deviation, and the lower minimum and maximum number of strokes.
